# Wnt-3a Induces Cytokine Release in Human Mast Cells

**DOI:** 10.3390/cells8111372

**Published:** 2019-11-01

**Authors:** Julia Tebroke, Joris E. Lieverse, Jesper Säfholm, Gunnar Schulte, Gunnar Nilsson, Elin Rönnberg

**Affiliations:** 1Division of Immunology and Allergy, Department of Medicine Solna, Karolinska Institutet, and Karolinska University Hospital, 171 64 Stockholm, Sweden; julia.tebroke.coe@googlemail.com (J.T.); joris.lieverse@gmail.com (J.E.L.); 2Experimental Asthma and Allergy Research, Institute of Environmental Medicine (IMM), Karolinska Institutet, 171 77 Stockholm, Sweden; jesper.safholm@ki.se; 3Section for Receptor Biology and Signaling, Department of Physiology and Pharmacology, Karolinska Institutet, 171 77 Stockholm, Sweden; gunnar.schulte@ki.se; 4Department of Medical Sciences, Uppsala University, 751 85 Uppsala, Sweden

**Keywords:** mast cells, Wnt signaling, Wnt-3a, Frizzled, IL-8, CCL8, asthma

## Abstract

Mast cells are well known for their detrimental effects in allergies and asthma, and Wnt signaling has recently been implicated in asthma and other airway diseases. However, it is not known if or how Wnts affect human mast cells. Since Wnt expression is elevated in individuals with asthma and is linked to a Th2 profile, we hypothesized that mast cells could be affected by Wnts in the context of asthma. We therefore sought to investigate the role of Wnt signaling in human mast cell development and activation. We first examined the expression of the 10 main Wnt receptors, Frizzled 1–10 (FZD_1–10_), and found expression of several FZDs in human mast cells. Treatment with purified recombinant Wnt-3a or Wnt-5a did not affect the proliferation or maturation of CD34^+^ progenitors into mast cells, as indicated by cellular expression of CD117 and FcεRI, activation by FcεRI crosslinking, and histamine and tryptase release. Furthermore, Wnt treatment did not change the phenotype from MC_T_ to MC_TC_, since MrgX2 expression, compound 48/80-mediated activation, and carboxypeptidase A3 content were not affected. However, Wnt-3a activated WNT/β-catenin signaling in mature human mast cells, as revealed by stabilization of β-catenin, upregulation of IL-8 and CCL8 mRNA expression, and release of IL-8 protein. Thus, our data suggest that Wnt-3a activation of mast cells could contribute to the recruitment of immune cells in conditions associated with increased Wnt-3a expression, such as asthma.

## 1. Introduction

Mast cells are important for surveillance of and responses to pathogens and cell injury but can also be detrimental to the host in the contexts of allergies, anaphylaxis, asthma, and other hypersensitivity reactions [1]. These cells are widely distributed in the body and are particularly numerous in areas exposed to the external environment, such as the lungs, skin, and gastrointestinal tract [1,2]. Mast cells originate form hematopoietic stem cells, circulate in the blood as mast cell progenitors, and undergo their final maturation in the tissues [2,3].

Human mast cells are classically divided into two different populations according to their expression of mast cell proteases, with mast cells of the MC_T_ class expressing tryptase only and those of the MC_TC_ subclass expressing tryptase, chymase, and carboxypeptidase A3 (CPA3). The MC_T_ subclass is mainly present on mucosal surfaces, such as those of the lungs and gut, whereas the MC_TC_ subclass dominates in connective tissues, such as the skin and intestinal submucosa [2,4].

The Wnt lipoglycoproteins form a family of 19 secreted ligands that are recognized by 10 Frizzled receptors (FZD_1–10_), G protein-coupled receptors that associate with various coreceptors [5]. Wnt ligands and their receptors participate in numerous possible ligand/receptor/coreceptor interactions and signal through different downstream pathways, which have been divided into three branches; these branches include the β-catenin-dependent or WNT/β-catenin pathway and an additional network of β-catenin-independent pathways, such as planar cell polarity (PCP)-like signaling pathways and G protein-dependent signaling pathways [5,6]. In addition to its many roles in embryonic development, tissue maintenance, and cell proliferation and differentiation, Wnt signaling has recently been implicated in critical regulation of inflammatory and structural responses in airway diseases and asthma (e.g., airway remodeling and airway smooth muscle proliferation). It has been described that Wnt ligands directly affect immune cells and seem to suppress the development of allergic airway diseases [7]. Additionally, the expression of Wnt-associated signaling molecules during airway development is linked to impaired lung function [8]. Furthermore, multiple Wnt genes (e.g., WNT3A, WNT5a, WNT6, and WNT10A) and the receptor FZD_5_ are positively correlated to a Th2 signature in the airways of humans with asthma [9].

Wnt signaling has been shown to have diverse effects on blood cell development [10]. It is downregulated during hematopoietic differentiation of embryonal murine stem cells and crucial for the maintenance of undifferentiated embryonic stem cells [11]. Furthermore, in human cells, Wnt/β-catenin signaling maintains a less differentiated hematopoietic stem cell phenotype in vitro [12,13]. In contrast, a recent study has shown that Wnt signaling promotes the terminal differentiation of murine mast cells in vitro via the WNT/β-catenin pathway. The receptors FZD_4_ and LRP5 have been found to be expressed in murine bone-marrow-derived mast cells (BMMCs) and peritoneal mast cells. Furthermore, Wnt-5a has been found to promote the maturation of murine BMMCs into a connective tissue phenotype [14].

Given the link between Wnt signaling and asthma and the roles of Wnts in murine mast cell development, we sought to investigate the role of Wnt signaling in human mast cell development and activation. Our results demonstrate the expression of FZDs and coreceptors on human lung mast cells and in vitro cultured mast cells and show that Wnts do not affect human mast cell development but do activate mature human mast cells and upregulate CCL8 (MCP-2) and CXCL8 (IL-8).

## 2. Materials and Methods

### 2.1. Ethical Approval

The local ethics committee approved the experiments involving human lung tissue, that is, (Regionala Etikprövningsnämnden Stockholm, 2010/181-31/2), the collection of lung tissue from patients undergoing lobectomies, and all patients provided informed consent, where six patients were included in the study. In accordance with Swedish legislation, ethics approval is not needed for the anonymous collection of cord blood and buffy coats because the samples cannot be traced to a specific person. Seven buffy coats (four for Figure 2 and three for Figures 3 and 4) and cord blood from 16 donors (Figures 5 and 6) were included in the study.

### 2.2. Cell Culture

Cord-blood-derived mast cells (CBMCs) were cultured as previously described [15]. Human lung mast cells were also obtained as previously described [16]. Briefly, human lung tissue was cut into small pieces and enzymatically digested for 45 min with DNase I and Collagenase. Thereafter, the tissue was mechanically disrupted by plunging through a syringe, the cells were washed, and debris was removed by 30% Percoll centrifugation. Pure human lung mast cells were obtained by fluorescence-activated cell sorting (FACS). Hematopoietic progenitors were isolated from the buffy coats of healthy donors. First, mononuclear cells were purified using Ficoll-Paque PLUS (GE Healthcare, Chicago, IL, USA), and this was followed by enrichment of CD34 progenitors with a CD34 MicroBead Kit (Miltenyi Biotec, Bergisch Gladbach, Germany). Isolated CD34 progenitors were cultured with complete Gibco StemPro-34 serum-free medium (Thermo Fisher Scientific, Waltham, MA, USA) with 2 mM L-glutamine (Sigma-Aldrich, St. Louis, MO, USA), 100 U/mL penicillin (Sigma-Aldrich), and 0.1 mg/mL streptomycin (Sigma-Aldrich). For short-term cultures of five days, the medium was supplemented with the recombinant human cytokines IL-3 (10 ng/mL; PeproTech, Rocky Hill, NJ, USA) and IL-6 (10 ng/mL; PeproTech), and for long-term culture, stem cell factor (SCF) (100 ng/mL, r-metHuSCF; Swedish Orphan Biovitrum, Stockholm, Sweden) was added. After week 1, the cells were cultured under the same conditions but without IL-3, and the media were exchanged weekly for a total of seven weeks. Enzyme cytochemical staining with Z-Gly-Pro-Arg-4-methoxy-b-naphthylamide substrate (Bachem, Bubendorf, Switzerland) was used to assess trypsin-like (tryptase) activity [17].

### 2.3. Treatment with Wnts and Mast Cell Activators

The cells were cultured under three different conditions: with 100 ng/mL recombinant human Wnt 3a or Wnt 5a (both R&D Systems, Minneapolis, MN, USA) or without Wnts. For long-term cultures, the Wnts were added weekly. For acute stimulation of mature CBMCs with Wnts (1–24 h), 300 ng/mL Wnt was added. For degranulation assays, 10 ng/mL IL-4 (PeproTech) was added four days prior and 1 µg/mL human IgE (Calbiochem, Minneapolis, MN, USA) was added one day prior to IgE–receptor crosslinking. The cells were crosslinked with 2 µg/mL anti-IgE antibody (Sigma-Aldrich). For activation via MrgX2, 6 μg/mL compound 48/80 (Sigma-Aldrich) was added, and the calcium ionophore A23187 (2 µM, Sigma-Aldrich) was used as a positive control for activation. The cells were incubated for 30 min at 37 °C, the supernatant was collected, and the cells were analyzed by flow cytometry.

### 2.4. Measurement of Mediator Release

Released histamine was measured using a histamine release test kit according to the manufacturer’s instructions (RefLab, Copenhagen, Denmark). Briefly, this test is based on the adsorption of histamine to fiberglass-coated microtiter plates. The fiberglass binds histamine with high affinity and selectivity. The plates were sent to RefLab, and released histamine was detected fluorometrically (with the o-phthalaldehyde (OPA) method) with a HISTAREADER™ 501-1. Tryptase and CPA3 activity in supernatants from activated mast cells was tested using specific chromogenic peptides. The supernatants were diluted 1 to 10 in PBS. Immediately before the assays were run on a Spectra Max iD3 (Molecular Devices), either Chromogenix S-2288™ (Diapharma, Bedford, MA, USA; for detection of trypsin-like activity/tryptase) or N-(4-methoxyphenylazoformyl)-Phe-OH· potassium salt (Bachem; for CPA3 analysis) was added to a final concentration of 0.3 mM. The absorbance was measured every minute for 30 min at 405 nm.

### 2.5. Flow Cytometry Analysis and Cell Sorting

The following stains/antibodies were used for surface staining: BD Horizon™ Fixable Viability Stain 450 (BD Biosciences, San Jose, CA, USA), CD45-V500 (clone HI30, BD Biosciences), CD14- APC-Cy7 (clone M5E2, BioLegend, San Diego, CA, USA), CD117-APC (clone 104D2, BD Biosciences), FcεRIα-PE and FcεRIα-FITC (clone AER-37 (CRA-1), BioLegend), CD34-Pe-Cy7 (clone 581, BD Biosciences), Integrin-β7-FITC (clone FIB504, eBioscience), MrgX2-PE (clone K125H4, BioLegend), and CD63-Pe-Cy7 (clone H5C6, BD Biosciences). To measure proliferation, the cells were labeled with CellTrace™ Far Red (Thermo Fisher Scientific) prior to treatment. Pure human lung mast cells were obtained by FACS of CD45^+^CD14^−^CD117^high^ cells using a FACSAria I instrument, flow cytometric analyses were performed with a FACSCanto II instrument (BD, Franklin Lakes, NJ, USA), and flow cytometry data analysis was performed with FlowJo software version 10 (FlowJo LLC, Ashland, OR, USA).

### 2.6. Western Blot Analysis

Cells (200,000 per sample) were lysed in Laemmli sample buffer (Bio-Rad Laboratories, Hercules, CA, USA), heated for 5 min at 95 °C, separated on Mini Protean TGX precast gels (Bio-Rad), and thereafter electroblotted onto nitrocellulose membranes (Bio-Rad). The membranes were blocked for 1 h with 5% BSA in PBS with 0.1% Tween-20. β-Catenin was detected using a β-catenin antibody (clone 14/Beta-Catenin, BD Biosciences) followed by HRP-linked anti-mouse IgG (GE Healthcare, Uppsala, Sweden). The Western blots were visualized by chemiluminescence (GE Healthcare) on a ChemiDoc Imaging System (Bio-Rad). The membranes were stripped using Re-Blot Plus Mild (Millipore) followed by reblocking and detection of β-Actin using an HRP-conjugated β-Actin antibody (clone C4, Santa Cruz Biotechnology, Dallas, TX, USA).

### 2.7. Quantitative PCR (qPCR) and RNA Sequencing (RNAseq)

For qPCR, RNA was extracted using an RNeasy Plus Mini Kit (Qiagen, Hilden, Germany), cDNA was prepared using an iScript cDNA synthesis kit (Bio-Rad), and qPCR was performed using TaqMan™ Gene Expression master mix (Thermo Fisher Scientific) on a CFX96 Real-Time System (Bio-Rad). The following primers were used: Hs02800695_m1 (HPRT1), Hs00174103_m1 (CXCL8), Hs04187715_m1 (CCL8), Hs00171147_m1 (CCL7), Hs01099660_g1 (CXCL5), Hs00234142_m1 (CCL3), and TaqMan Array Human WNT Pathway (all from Thermo Fisher Scientific). The results were calculated using the ΔCT method, and the CT values were normalized to the housekeeping gene GAPDH or HRPT1. For RNAseq, RNA was extracted using Arcturus PicoPure RNA Isolation Kit (Thermo Fisher Scientific).

### 2.8. Statistical Analysis

The data are shown as the mean with the standard error of the mean (SEM). Statistical analyses were performed with GraphPad Prism software version 7.0b (San Diego, CA, USA). One-way or two-way ANOVA with Bonferroni’s post hoc test was performed (* *p* < 0.05; ** *p* < 0.01; *** *p* < 0.001; **** *p* < 0.0001).

## 3. Results

### 3.1. Human Mast Cells Express FZDs

We first investigated the mRNA expression of FZD_1–10_ and their coreceptors in in vitro cultured CBMCs and human lung mast cells by qPCR. We found detectable expression of several FZDs in CBMCs (Figure 1A) and human lung mast cells (Appendix A). The expression of FZDs in human lung mast cells was also confirmed using RNA sequencing (Table 1). In addition, we examined the expression of FZDs in human skin mast cells in the online depository of FANTOM5 and they also expressed FZDs (Appendix A) [18]. Both CBMCs and lung mast cells also expressed the relevant intracellular scaffold proteins Disheveled (DVL) 1, 2, and 3 and the coreceptors LRP5-6 (Figure 1B, Appendix A, Table 1). We also measured the expression of the 19 WNTs and found that both lung mast cells (Appendix A and Table 1) and CBMCs (Figure 1C) expressed primarily WNT11, implying the existence of a possible autocrine loop. Furthermore, we analyzed human lung tissue for expression of WNTs and found that several WNTs were abundantly expressed (Appendix A). In summary, human mast cells express the required receptors for functional responses to autocrine or paracrine stimulation with Wnts and should thus recognize and react to Wnts expressed in the lungs.

### 3.2. Wnts Do Not Affect Early Mast Cell Development

Wnt signaling is known to affect cell differentiation and proliferation; thus, we investigated the effects of Wnt-3a and Wnt-5a, whose expression is increased in Th2-asthma [9], on early human mast cell development. CD34^+^ progenitors were enriched from human peripheral blood and cultured under conditions that promote mast cell development [19] with or without Wnt-3a or Wnt-5a. After five days of culture, the percentages of mast cell progenitors/early mast cells did not differ under the different conditions, nor did the Wnts affect the CD117^−^FcεRI^+^ percentages, a gate where basophils are located (Figure 2A–C). The decrease in CD34 after five days of culture was similar in the gated mast cell progenitors/early mast cells with and without Wnts (Figure 2D), but the decrease in integrin β7 was statistically significant only in the Wnt-treated groups (Figure 2E). Neither the size nor the granularity of the mast cell progenitors/early mast cells (forward scatter (FSC) and side scatter (SSC)) was affected by Wnt treatment (Figure 2F,G).

### 3.3. Wnts Do Not Affect Mast Cell Maturation

We next investigated the effects of the Wnts on the maturation of CD34^+^ blood mast cell progenitors into mature mast cells by adding Wnt-3a and Wnt-5a every week during the culture period of seven weeks. Wnt treatment affected neither the total cell numbers during the culture period (Figure 3A) nor the percentages of tryptase-positive mast cells (Figure 3B,C) or CD117^+^FcεRI^+^ cells (Figure 3D,E) after seven weeks of culture. We then investigated the phenotypes of the in vitro developed mast cells at week 7 and found no effect on the expression of the receptors CD117, FcεRI, and MrgX2 (data not shown) or on the size and granularity of the cells (FSC and SSC) (Figure 3F,G).

To examine if treatment with Wnt-3a or Wnt-5a during seven weeks of culture could affect mast cell reactivity, the mature mast cells were activated by crosslinking of the FcεRI receptor with anti-IgE, treatment with compound 48/80 (to activate MrgX2), and treatment with the calcium ionophore A23187. Compound 48/80 did not cause any significant degranulation of the cells, as determined by CD63 expression on the cell surface, but anti-IgE and A23187 both potently activated the mast cells (Figure 4A,B). However, there were no effects of culturing with Wnts on mast cell activation (Figure 4B–C). Since it has been reported that Wnt-5a affects mouse mast cell histamine, tryptase, and CPA3 content, we assessed histamine (Figure 4C), tryptase (Figure 4D), and CPA3 activity (largely undetectable; data not shown) in the supernatant; however, there was no effect of Wnt treatment.

### 3.4. Wnt-3a Activates Mature Mast Cells and Causes IL-8 Release

In the WNT/β-catenin signaling pathway, Wnt stimulation leads to stabilization and accumulation of β-catenin protein, which acts as a transcriptional coactivator after translocation into the nucleus [20]. Therefore, we examined the β-catenin content after addition of Wnt-3a or Wnt-5a to mature mast cells (CBMCs) by Western blot analysis. Wnt-3a, but not Wnt-5a, caused a robust induction of β-catenin (Figure 5A). To examine the possible effects of this induction, we investigated proliferation, degranulation, and cytokine release. The Wnts did not induce degranulation, nor did two hours of pretreatment with the Wnts affect IgE-mediated degranulation (Figure 5B). Similarly, Wnt treatment did not affect proliferation of mature CBMCs (Figure 5C).

Since Wnt signaling has been shown to induce cytokine release from other immune cells [21,22,23], we next examined cytokine secretion in response to Wnt-3a. Supernatants from CBMCs stimulated for 24 h with Wnt-3a were analyzed on an Olink proteomics inflammation panel, including 92 inflammation-related proteins. The data obtained indicated that a number of chemokines were released in response to Wnt-3a (Supp. Figure 2). A few candidates were chosen for further investigation: IL-8 (CXCL8), CCL3 (MIP-1α), CCL7 (MCP-3), CCL8 (MCP-2), and CXCL5. A time-course experiment showed that induction of IL-8 mRNA expression by Wnt-3a peaked early after stimulation (Figure 6A), and the two-hour time point was chosen for further investigations. Analyses of CBMCs from seven different donors treated with Wnt-3a showed that IL-8 and CCL8 mRNA expression (Figure 6B,E) was induced by Wnt-3a, while there was no significant change in CCL3, CCL7, or CXCL5 expression (Figure 6C,D,F). Wnt-5a did not induce expression of any of the chemokines. The Wnt-3a-induced expression and release of IL-8 protein were confirmed by ELISA (Figure 6G).

## 4. Discussion

Wnt signaling has been shown to play an important role in airway pathologies, such as asthma [7]. We show here that mature human mast cells, including primary lung mast cells, express FZDs, the central scaffold proteins DVL1-3, and the coreceptors LRP5 and LRP6, indicating that they have the molecular machinery to respond to Wnts (Figure 1A–C, Appendix A). Western blots of Wnt-stimulated CBMCs showed that Wnt-3a activated the WNT/β-catenin pathway (Figure 5A). Wnt treatment, however, did not cause a classical degranulation response, as no histamine was released, nor did it influence mast cell degranulation by FcεRI crosslinking (Figure 5B). Other compounds, such as toll-like receptor agonists, have been shown to induce mast cell activation and cytokine production without signs of classical degranulation [24]. In addition, Wnts have previously been shown to induce cytokine expression in other immune cells [21,22,23]. Using an Olink proteomics inflammation panel screen, we found indications that certain chemokines were released in response to Wnt-3a (Appendix A); in addition, upregulation of IL-8 and CCL8 mRNA was confirmed by qPCR, and increased release of IL-8 was confirmed by ELISA (Figure 6A,B,E,G). Expression of Wnt-3a in the lung has been shown to correlate with a Th2 signature in individuals with asthma [9]; therefore, in the context of Th2 inflammation in the lung, Wnt-3a activation of mast cells to release chemokines and subsequent recruitment of other immune cells could contribute to the pathology.

Wnt-5a have previously been shown to induce maturation of murine mast cells [14]; we could not, however, confirm the corresponding effect in human mast cells. Stimulation with Wnt-3a or Wnt-5a did not significantly influence the maturation, differentiation, or proliferation of human mast cell progenitors in vitro (Figure 2 and Figure 3). Murine mast cells are also divided into two major subclasses, mucosal mast cells (MMCs) and connective-tissue-type mast cells (CTMCs), depending on the expression of mast cell proteases [2,4]. Yamaguchi showed that Wnt signaling induced the formation of more mature CTMC-like mouse mast cells with elevated mRNA expression of histidine decarboxylase (HDC, the rate-limiting enzyme in histamine biosynthesis), mMCP-5, and CPA3 as well as elevated tryptase and CPA3 activity. The Wnt-treated mast cells also degranulated more strongly than non-Wnt-treated mast cells in response to compound 48/80 [14]. In our experiments, compared to the control cells, the mast cells generated in the presence of Wnts had equal levels of CD117 and FcεRI, degranulated to the same extent in response to treatment with FcεRI and the calcium ionophore A23187, and released equal levels of histamine and tryptase. Collectively, our findings indicate that the Wnts had no effect on the maturation level of the cells. In humans, the MC_TC_ subtype is more abundant in connective tissues such as the skin and expresses higher levels of chymase, CPA3, and the MrgX2 receptor, which is activated by basic substances such as compound 48/80, than the MC_T_ subtype [25]. To investigate if the mast cells developed in the presence of Wnt were of the MC_TC_ type, we measured surface MrgX2 expression and degranulation in response to compound 48/80 as well as CPA3 activity. The developed mast cells had very low expression of MrgX2 (data not shown) and did not degranulate in response to compound 48/80, and neither Wnt-3a nor Wnt-5a influenced these characteristics (Figure 4B). CPA3 activity was detectable only in the supernatant from one donor, and there was no significant change upon the addition of Wnts (data not shown). Together, these results show that in contrast to the case in mouse mast cells, in human mast cells, Wnt-3a or Wnt-5a does not induce a more mature, connective-tissue-associated MC_TC_ phenotype. We can only speculate as to the reason behind the discrepancies in Wnt signaling between mouse and human mast cells, where one possibility is the expression of FZD_4_ by mouse mast cells [14], whereas CBMCs showed a very low expression of FZD_4_ and it was not detectable in human lung mast cells (Figure 1A, Appendix A).

So far, it is poorly understood which out of the 19 mammalian Wnts interacts with which of the 10 paralogs of FZDs and what determines selectivity and pathway initiation. Immunoprecipitation studies have shown that Wnt-3a interacts with FZD_1, 3, 5–8_ and Wnt-5a with FZD_1–3, 5–6_ [26]. Wnt-3a and Wnt-5Aa are intrinsically different. Wnt-3a is a strong activator of the WNT/β-catenin pathway [27], whereas Wnt-5a generally signals in a β-catenin-independent manner regulating planar-cell-polarity-like signaling and activation of heterotrimeric G proteins [28]. However, this does not mean that Wnt-3a solely acts through β-catenin-dependent pathways. In primary mouse microglia, for example, Wnt-3a activates the WNT/β-catenin pathway in parallel to G protein-mediated mitogen-activated protein kinase signaling [29]. Thus, future work will be required to dissect the underlying signaling pathways initiated by Wnt-3a in mast cells.

In summary, we have shown that mast cells express FZDs, DVL1-3, and LRP5/6. Wnt-3a and Wnt-5a do not influence human mast cell differentiation, maturation, or proliferation, but Wnt-3a activates mature mast cells to produce the chemokines IL-8 and CCL8. This activation could contribute to the recruitment of immune cells in conditions associated with increased Wnt-3a expression, such as asthma. Inhibitors targeting Wnt signaling is under evaluation for the treatment of idiopathic pulmonary fibrosis [30]. Our results and recent findings linking Wnt signaling to asthma point to the possibility that asthma patients could also benefit from such inhibitors [31]. However, considering the many functions of Wnt signaling, caution needs to be taken when targeting this pathway.

## Figures and Tables

**Figure 1 cells-08-01372-f001:**
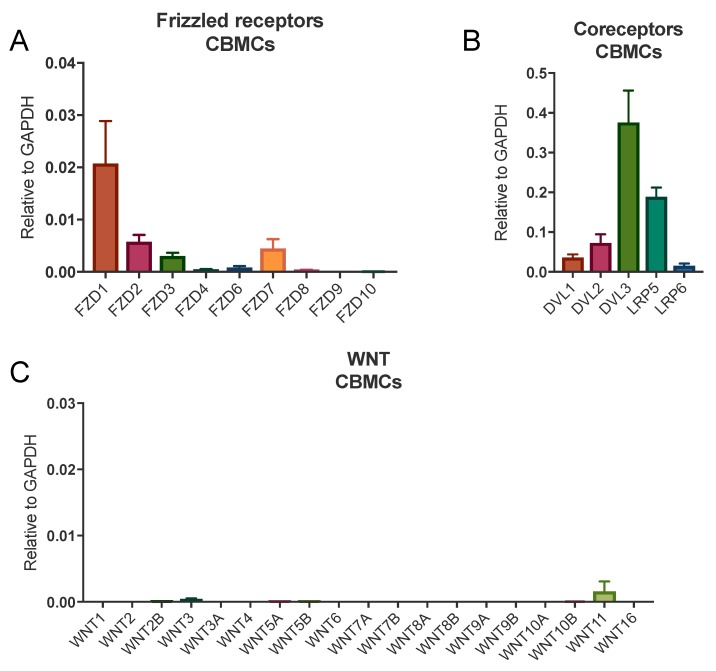
mRNA expression of components of the Wnt signaling system in human mast cells. mRNA was extracted from human cultured CBMCs and qPCR was performed for FZD_1–10_ (**A**), DVL1-3 and LRP5/6 (**B**), and all 19 WNTs (**C**) using a Human WNT Pathway TaqMan Array. *n* = 3, means with SEMs.

**Figure 2 cells-08-01372-f002:**
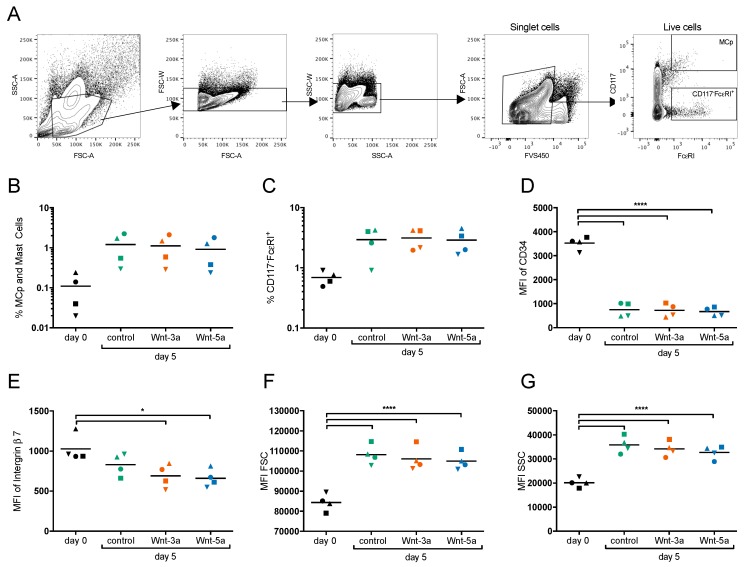
Stimulation with purified recombinant Wnt-3a and Wnt-5a did not influence early mast cell progenitor development. CD34^+^ cells were enriched from buffy coats and cultured for five days with or without 100 ng/mL Wnt-3a or Wnt-5a under conditions that promote mast cell development. The cells were gated for single cells/live cells, and mast cells/mast cell progenitors (MCps) were gated as CD117^high^FcεRI^+^ and CD117^−^FcεRI^+^ for basophils (**A**). Frequencies of early mast cells and mast cell progenitors (**B**). Frequency of CD117^−^FcεRI^+^ (**C**). CD34 expression (**D**), Integrin β7 expression (**E**), FSC (**F**) and SSC (**G**) for the gated mast cell progenitors/early mast cells. *n* = 4; each symbol represents an individual culture. * *p* < 0.05; **** *p* < 0.0001.

**Figure 3 cells-08-01372-f003:**
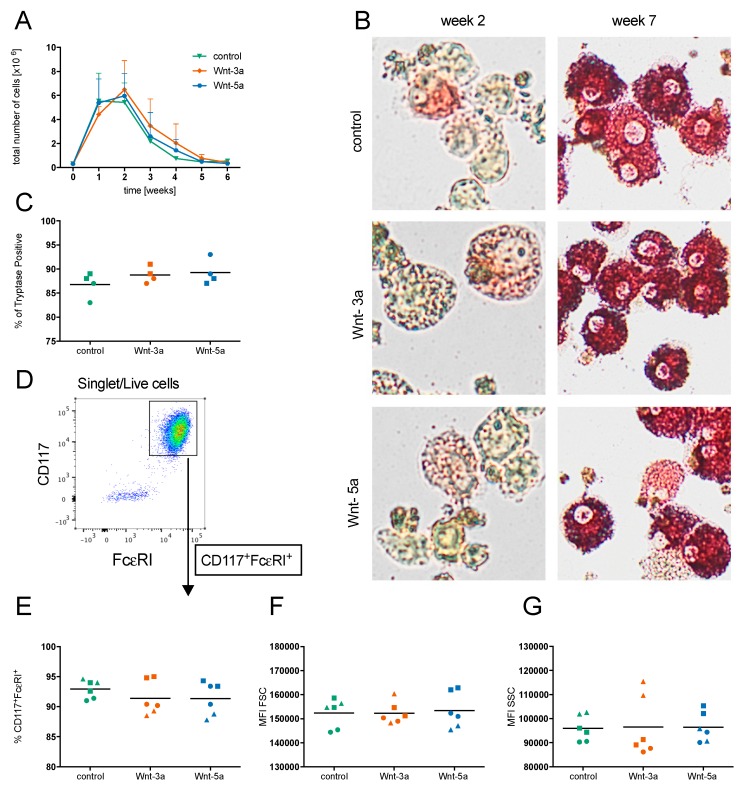
Stimulation with purified recombinant WNT does not influence mast cell maturation. CD34^+^ cells enriched from buffy coats were cultured for seven weeks under conditions that promote mast cell development, with weekly addition of 100 ng/mL Wnt-3a or Wnt-5a. The total number of cells during the culture period was quantified as the means with SEMs (**A**). The cells were stained for tryptase activity at week 2 and week 7 (**B**), and the percentages of tryptase-positive cells at week 7 were quantified (**C**). The cells were analyzed by flow cytometry; representative gating of developed mast cells at week 7 is shown in (**D**), and quantification of the gated CD117^high^FcεRI^high^ mast cells is shown in (**E**). Mean fluorescence intensity (MFI) of the FSC (**F**) and SSC (**G**) of the gated mast cells. Cells from three individual donors were analyzed in duplicate (*n* = 3), and each symbol represents an individual donor.

**Figure 4 cells-08-01372-f004:**
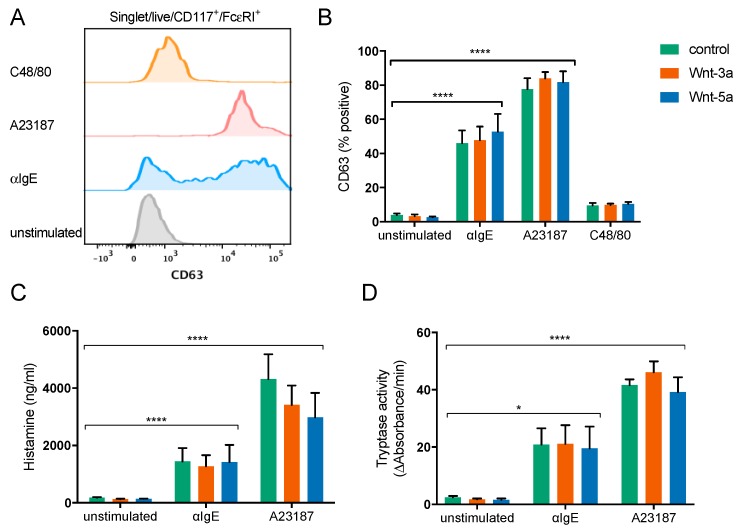
Mast cells cultured in the presence of purified recombinant Wnts show equal activation and release of mediators. CD34^+^ cells enriched from buffy coats were cultured for seven weeks under conditions that promote mast cell development, with weekly addition of 100 ng/mL Wnt-3a, Wnt-5a, or no Wnts. The cells were thereafter activated with αIgE, A23187 or compound 48/80 (C48/80) and activation was measured by CD63 surface expression using flow cytometry. A representative histogram is shown (**A**). The percentages of CD63^+^ cells were quantified (**B**), and histamine (**C**) and tryptase activity (**D**) was measured in the supernatant. Cells from three individual donors were analyzed in duplicate; *n* = 3, means with SEMs. * *p* < 0.05; **** *p* < 0.0001.

**Figure 5 cells-08-01372-f005:**
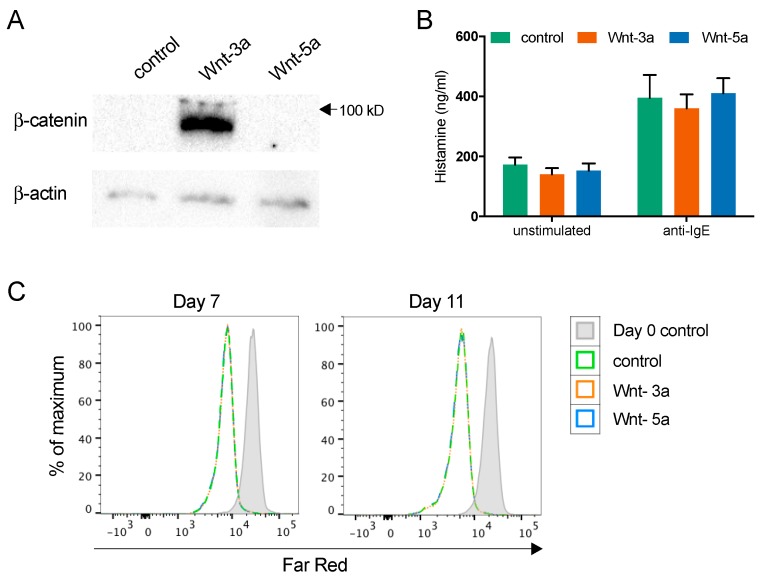
Wnt-3a activates the WNT/β-catenin pathway in mature human mast cells. CBMCs were treated for 2 h with or without 300 ng/mL Wnt-3a or Wnt-5a, and Western blot analysis of β-catenin and β-actin was performed on the cell lysates (**A**). CBMCs were treated for 2 h with 300 ng/mL Wnts and thereafter activated with anti-IgE. The supernatant was collected after 30 min, and histamine release was measured; *n* = 6, means with SEMs (**B**). CBMCs were labeled with the cell proliferation dye CellTrace Far Red, cultured with or without 100 ng/mL Wnt-3a or Wnt-5a for 7 or 11 days and analyzed by flow cytometry. A representative histogram of three independent cultures for each of the different conditions is shown (**C**).

**Figure 6 cells-08-01372-f006:**
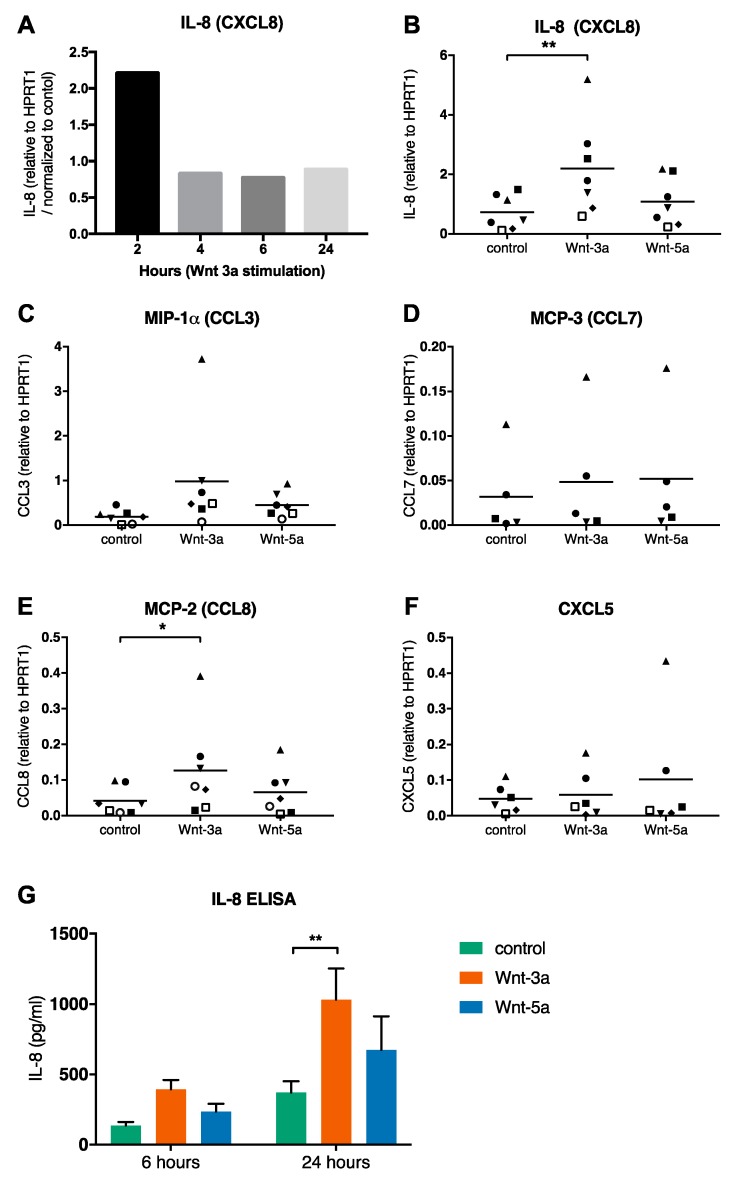
Wnt-3a upregulates IL-8 and CCL8 mRNA and induces IL-8 protein secretion. CBMCs were treated for 2 h (unless otherwise stated) with or without 300 ng/mL Wnt-3a or Wnt-5a, and qPCR was performed for IL-8 (**A**,**B**), CCL-3 (**C**), CCL7 (**D**), CCL8 (**E**), and CXCL5 (**F**). IL-8 released into the supernatant was measured by ELISA; *n* = 6, means with SEMs (**G**). Each symbol (**B**–**F**) represents data from an individual cord blood donor, *n* = 5–7. * *p* < 0.05; ** *p* < 0.01;

**Table 1 cells-08-01372-t001:** mRNA expression of the Wnt signaling system in human lung mast cells. mRNA was extracted from sorted human lung mast cells and RNAseq was performed. DESeq2 normalized counts of FZDs, DVL1-3, LRP5/6, and all 19 WNTs are shown.

Gene Name	Donor 1	Donor 2	Donor 3	Average
FZD1	304	176	151	210
FZD2	58	8	8	25
FZD3	369	137	301	269
FZD4	4	6	2	4
FZD5	75	96	23	65
FZD6	16	8	16	13
FZD7	225	59	42	109
FZD8	12	2	2	5
FZD9	2	0	0	1
FZD10	0	0	0	0
DVL1	43	33	48	41
DVL2	193	156	206	185
DVL3	723	331	408	488
LRP5	35	16	18	23
LRP6	82	104	81	89
WNT1	2	0	0	1
WNT2	2	2	0	1
WNT2B	17	23	12	17
WNT3	1	2	5	2
WNT3A	0	0	0	0
WNT4	1	5	1	2
WNT5A	7	2	0	3
WNT5B	6	2	1	3
WNT6	3	0	0	1
WNT7A	1	0	3	1
WNT7B	0	0	2	1
WNT8A	0	0	0	0
WNT8B	0	0	0	0
WNT9A	0	0	0	0
WNT9B	0	0	0	0
WNT10A	10	0	2	4
WNT10B	1	5	2	2
WNT11	145	131	26	101
WNT16	1	0	5	2

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
