# Peer review of "Wnt-3a Induces Cytokine Release in Human Mast Cells"

_cells, 2019, doi:10.3390/cells8111372_

Round 1

Reviewer 1 Report

My concerns have been addressed and I now recommend acceptance of the manuscript.

Reviewer 2 Report

no further comments

Reviewer 3 Report

Dear authors, 

Thank you for the effort to modify the manuscript according to my suggestions. However, I can not recommend the publication of manuscript that analyzed blood from a donor without approvement of an Ethical Committee. The Standards and operational guidance for ethics review of health-related research with human participants from World Healthy Organization describe that "research ethics committees may review different types of research studies, including, but not limited to, the following: • clinical trials • research on medical records or other personal information • research on stored samples (available on https://apps.who.int/iris/bitstream/handle/10665/44783/9789241502948_eng.pdf;jsessionid=21D7B7973A1767AF53CF9902A6DEEA3C?sequence=1)

In addition, WHO describes that "all research involving human beings should be reviewed by an ethics committee to ensure that the appropriate ethical standards are being upheld."

Moreover, regarding data of Figure 1S: how mRNA from pooled samples from 3 donors result in n=4?  Since the authors used pooled samples, the n is still one, correct?